# Potential of Mesenchymal Stromal Cell-Derived Extracellular Vesicles as Natural Nanocarriers: Concise Review

**DOI:** 10.3390/pharmaceutics15020558

**Published:** 2023-02-07

**Authors:** Florian Draguet, Cyril Bouland, Nathan Dubois, Dominique Bron, Nathalie Meuleman, Basile Stamatopoulos, Laurence Lagneaux

**Affiliations:** 1Laboratory of Clinical Cell Therapy (LCCT), Jules Bordet Institute, Université Libre de Bruxelles (ULB), 90 Rue Meylemeersch, 1070 Brussels, Belgium; 2Department of Stomatology and Maxillofacial Surgery, Saint-Pierre Hospital, 322 Rue Haute, 1000 Brussels, Belgium; 3Department of Maxillofacial and Reconstructive Surgery, Grand Hôpital de Charleroi, 3 Grand’Rue, 6000 Charleroi, Belgium; 4Department of Haematology, Jules Bordet Institute, Université Libre de Bruxelles (ULB), 90 Rue Meylemeersch, 1070 Brussels, Belgium; 5Medicine Faculty, Université Libre de Bruxelles (ULB), Route de Lennik 808, 1070 Brussels, Belgium

**Keywords:** extracellular vesicles (EVs), exosomes, mesenchymal stromal cells (MSCs), drug delivery system (DDS), diseases

## Abstract

Intercellular communication, through direct and indirect cell contact, is mandatory in multicellular organisms. These last years, the microenvironment, and in particular, transfer by extracellular vesicles (EVs), has emerged as a new communication mechanism. Different biological fluids and cell types are common sources of EVs. EVs play different roles, acting as signalosomes, biomarkers, and therapeutic agents. As therapeutic agents, MSC-derived EVs display numerous advantages: they are biocompatible, non-immunogenic, and stable in circulation, and they are able to cross biological barriers. Furthermore, EVs have a great potential for drug delivery. Different EV isolation protocols and loading methods have been tested and compared. Published and ongoing clinical trials, and numerous preclinical studies indicate that EVs are safe and well tolerated. Moreover, the latest studies suggest their applications as nanocarriers. The current review will describe the potential for MSC-derived EVs as drug delivery systems (DDS) in disease treatment, and their advantages. Thereafter, we will outline the different EV isolation methods and loading techniques, and analyze relevant preclinical studies. Finally, we will describe ongoing and published clinical studies. These elements will outline the benefits of MSC-derived EV DDS over several aspects.

## 1. Introduction

Intercellular communication is essential in multicellular organisms. It occurs through direct and indirect cell contact. In these last years, the microenvironment, and in particular, the transfer of extracellular vesicles (EVs), was highlighted as a new communication mechanism. Different biological fluids and cell types are common sources of EVs [1,2,3]. MSC-derived EVs display several advantages: biocompatibility, immunological inertness, and the ability to cross biological barriers [4]. In recent years, different roles have been suggested for EVs as signalosomes, biomarkers, and therapeutic agents [5,6]. EVs have been used in several conditions or diseases: cancers (brain, breast, lung, colorectal, and liver cancers and lymphoma), cardiovascular diseases (infarction and stroke), neurological diseases (Parkinson’s and Alzheimer’s), inflammatory diseases (arthritis and allergic cutaneous contact dermatitis), infectious diseases (HIV-1 and tuberculosis), obesity, diabetes, and others (kidney disease, liver disease, muscular disease, cutaneous wounds, and immunomodulation) [7]. Furthermore, EVs have emerged as a novel drug delivery vehicle [5,6,8]. Different approaches have been tested to isolate and to purify EVs, such as ultracentrifugation, ultrafiltration, size-exclusion chromatography, and immunoaffinity. Many different loading methods have been developed to promote drug delivery by EVs, and these include endogenous or exogenous approaches. The endogenous approach involves the modification of parental cells through transfection [9], lipofection [10], or coincubation with a drug before the purification of these modified MSC-derived EVs [11]. Through exogenous methods, drugs or molecules are loaded in EVs after their purification. The most commonly used methods are coincubation with drug (passive loading) [10] or active loading via electroporation [12], sonication [13], freeze/thaw cycles [14], or permeabilization with saponin and extrusion [15].

Completed and ongoing clinical trials, as well as numerous preclinical studies, indicate that EVs are safe and well tolerated. Moreover, the latest studies suggest their application as nanocarriers [16,17].

In this review, we will first describe the role of MSC-derived EVs in disease treatment, and their potential advantages. Thereafter, we will outline the different EV isolation methods and loading techniques, and analyze relevant preclinical studies. Finally, we will describe ongoing and published clinical studies. These elements will outline the benefits of MSC-derived EVs as drug delivery systems in several aspects.

## 2. Mesenchymal Stromal Cells (MSCs)

MSCs are multipotent adult stem cells with multilineage differentiation potential [18]. The International Society for Cellular Therapy (ISCT) defined MSC as the association of the following criteria: adherence to plastic, specific surface antigen (Ag) expression, and osteoblasts, chondrocytes, and adipocytes differentiation capacity [19]. Different MSC sources have been highlighted (Figure 1): bone marrow (BM), skeletal muscle tissue, adipose tissue (AT), synovial membranes, saphenous veins, dental pulp, periodontal ligaments, cervical tissue, Wharton’s jelly umbilical cords, umbilical cord blood, amniotic fluid, and placenta [18,20].

MSCs display differentiation, proliferation, immunomodulatory, and trophic properties, which have been harnessed in clinical applications [21]. Recently, it has been proposed that MSCs might exert their therapeutic effects through paracrine activity, through the secretion of biologically active cargo [22,23]. These active molecules comprise not only soluble factors, but also EVs, which have recently emerged as a cell–cell communication mechanism [23,24,25]. MSC-derived EVs have demonstrated benefits for the management of different pathological conditions. Recently, loading MSC-derived EVs with defined cargos such as miRNAs has been suggested to be a promising strategy for the treatment of different diseases [26]. For these reasons, it was important to describe the potential of MSC-derived EVs as drug delivery systems (DDS) in disease treatment and their advantages.

## 3. Mesenchymal Stromal Cell-Derived Extracellular Vesicles (MSC-Derived EVs)

Described 50 years ago as “platelet dust” [27], EVs were long considered as cellular debris. In the late 1990s, it was suggested that EVs could be relevant mediators of intercellular communication [28,29]. Today, they are known to play significant roles in complex biological processes: tumourigenesis [30], immunomodulation [25], anticancer therapy by acting as a nanocarrier of cytotoxic drugs (i.e., paclitaxel) [31], the suppression of apoptosis; and the stimulation of cell proliferation [32,33], angiogenesis [34], inflammation [35], and homeostasis [36].

EVs are phospholipid bilayer-enclosed vesicles [37]. They are divided into different groups according to their size (“small EVs” (sEVs) and “medium/large EVs” (m/lEVs), with ranges defined as <100 nm or <200 nm [small] or >200 nm [large and/or medium]) or density (low, medium, or high) [38]. EV cargo contains proteins, lipids, DNA, RNA, mRNA, and miRNA [1,2,39]. Their composition reflects the parental cell status at the time of production [40]. EVs have been isolated from different cell sources and biological fluids. Numerous cell types release EVs: MSC, T cells, B cells, dendritic cells, etc. [3,10]. They are positive for CD29, CD44, CD73, and CD105, similar to MSCs, and a variety of proteins are bound to EV membranes or are present in their intraluminal space. These include heat shock proteins (Hsp70 and Hsp90), lysosomal-associated membrane proteins (Lamp2a and Lamp2b), cytoskeletal proteins (actin, tubulin, and actinin-4), trafficking proteins (TSG101 and Alix), integrins, proteoglycans, tetraspanins (CD9, CD37, CD53, CD63, and CD81), and cytokines/interleukins [37]. By analyzing datasets of published MSC-derived EV proteomics, a specific protein signature was reported with 22 members and was found to be involved in functions as cell adhesion and integrin–receptor interaction [41]. Recently, the presence of 591 proteins and 604 microRNAs has been observed in adipose tissue MSC-derived EVs implicated in the binding functions, as well as signal transduction and gene silencing [42]. EVs are also enriched in lipids such as ceramides, cholesterol, and sphingomyelin, which promote vesicle release and play important roles in cell communication. These molecules affect the sorting of contents, secretion, structure, and EV signaling [43]. Importantly, the variations in lipid species determine the EV physiochemical properties, of which the zeta potential is negative, due to their membrane negatively charged lipids [44].

MSC therapies have long been considered to repair the affected structure and function of tissues through direct cell replacement. In vivo, MSCs migrate to injured sites. However, most of the engrafted MSCs are lost. Their immunomodulating action is considered to play a major role through the release of trophic factors. Nevertheless, studies have demonstrated that MSC culture medium produced a similar therapeutic effect to MSC therapy in retinal ischemia [45], or in diabetes mellitus [46]. Subsequent studies have highlighted the presence of EVs in the culture medium [47]. Recently, it has been suggested that MSCs might exert their therapeutic effects mainly through secreted extracellular factors [22]. As EVs are involved in cell–cell communication, it is hypothesized that EVs mediate the paracrine effects of MSCs [48]. MSC-derived EVs have been attributed to have MSC functions: facilitating heart repair [49], modulating immune responses [50], promoting bone healing [51], and acting as drug delivery carriers [52]. In addition, EV therapy displays other advantages over cell-based therapies in regenerative medicine [4,53]: (1) the cargo delivery of favorable miRNAs responsible for promoting angiogenesis, fibrosis, and cell proliferation; (2) the potential for “off the shelf” availability and for repetitive transplantation; (3) cell-free biological products that may be utilized as drug carrier systems in the pharmaceutical industry; and (4) reduced immunogenicity, which allows for allogeneic transplantation. In recent years, different roles have been attributed to EVs, such as signalosomes, biomarkers, and therapeutic agents [5,6]. Furthermore, EVs have emerged as a novel drug delivery vehicle [5,6].

## 4. Mesenchymal Stromal Cell-Derived Extracellular Vesicles (MSC-Derived EVs) for Drug Delivery Systems (DDS)

Several types of DDS have been considered for drug-targeting applications. Synthetic lipid nanoparticles (liposomes) are the most biocompatible and the least toxic artificial systems. They are composed of phospholipids and cholesterol, which are both components of cell membranes. Liposomes can entrap drugs in both aqueous and lipid phases, and thus deliver hydrophilic and hydrophobic drugs. They can load multiple drugs to increase drug delivery, and consequently, potentially reduce toxicity and increase the treatment effectiveness [54]. Liposomes can be endowed with specific targeting ligands to enhance the accumulation and intracellular uptake into target cells expressing the specific receptor [55]. However, these nanoparticles suffer from poor biocompatibility and biodegradability. In addition, immunogenicity limits their therapeutic applicability. EV-based DDS could resolve these drawbacks.

In contrast to liposomes, EVs share the lipid asymmetry of the parent cells, allowing for optimal interaction with their target cells [56]. Different studies have reported that EVs may be taken up more efficiently into target cells than the liposomes, leading to an enhanced delivery of the cargo contained in EVs [57,58].

EV-based DDS present multiple advantages (Figure 2): (1) their structure is composed of an aqueous core and a rich lipid bilayer surface structure, allowing for the compartmentalization and solubilization of both hydrophilic and lipophilic agents [59]. (2) EVs carry various biomolecules, such as proteins, lipids, and DNA and RNA species, depending on the producer cell types; the surface structure consists of fatty acids, high concentrations of cholesterol, sphingomyelin, ceramides, and other lipids; and interestingly, this surface also contains proteins that are implicated in adhesion, such as tetraspanins and αβ integrins, conferring on EVs an endogenous homing and targeting capacity [60]. (3) EVs are considered to be non-immunogenic, with a lower risk of allogeneic immune rejection from the host [61]. (4) Their surface composition can be modified through different engineering approaches [62,63]. (5) EVs can efficiently cross biological tissue, cellular and intracellular barriers (i.e., the gastrointestinal barrier and blood–brain barrier), and induce functional changes in the target cell [64]. Moreover, EVs have fewer off-target effects, due to the natural tendency to act on specific target cells.

EVs can also be produced by plant cells (PEVs) [65], or by bacteria (BEVs) [66] and fungi (FEVs) [67], and they contain bioactive molecules, displaying multiple functions. These EVs can deliver exogenous and endogenous agents to mammalian cells in the majority of organs, and they have also a great potential to become novel drug delivery systems. As human EVs, they display advantageous properties such as low immunogenicity, tissue-specific targeting, safety, negative zeta potential, and the ability to load many biomolecules [68]. However, the therapeutic potential of these EVs is still in its infancy, due to the absence of a comprehensive understanding of the biogenesis mechanism, internalization and packaging processes, cargo identification, and the comparison with liposomes-based methods [69].

Due to their cell-based biological structures and functions, EVs represent an ideal natural material for the development of nanomedicine. However, there are different challenges to face before any clinical application of EV-based drug delivery systems. Indeed, EV preparations are highly heterogeneous due to the difficult purification of a specific EV population. Moreover, the isolation and purification methods are not uniform, limiting standardization.

## 5. The Isolation of Extracellular Vesicles

The process of large-scale EV production includes the expansion of MSCs, the collection of conditioned medium, and the isolation of EVs. Numerous EV isolation methods have been described, from differential ultracentrifugation (UC) to immuno-isolation by different surface molecules through density gradients, polymer-based precipitation, microfiltration, and size-exclusion-based chromatography [70] (Table 1). Differential UC is the most commonly used method. The process is based on the separation of particles according to their buoyant density. This procedure includes several substeps: centrifugation at 300–400× *g* for 10 min to sediment cells, at 2000× *g* to remove cell debris, and at 10,000× *g* to remove the aggregates and apoptotic bodies. Thereafter, the EV pellet is obtained via UC (100,000–200,000× *g* for 2 h). Filtration can replace the low-speed centrifugal steps for the large-scale preparation of exosomes in specific cases [71]. The EV isolation efficiency after differential UC depends on many factors: acceleration, rotor type, and sample viscosity. Sucrose density gradients (sucrose, iohexol, and iodixanol) and UC can be applied to increase the efficiency of particle separation to obtain highly purified EVs [72]. However, both methods are expensive and time-consuming. Moreover, EV aggregation and rupture due to high shear forces have been reported [73].

Polymer-based precipitation is another isolation method. The method is based on EV precipitation in polymer solutions, due to changes in EV solubility and aggregation. The reagents used for polymer-based exosome isolation mainly include protamine, acetate, protein organic solvent precipitation (PROSPR), and polyethylene-glycol (PEG). PEG is the most commonly used polymer [74]. Dash et al. suggested that the PEG-based approaches display high stability and good quality [75]. This operation is simple, fast, and suitable for large-volume samples, and it preserves the bioactivity of isolated EVs. However, potential contamination with copurifying protein aggregates or residual polymers isolated with EVs may occur. The EV size is comparable between the UC and precipitation methods. Nevertheless, the EV count is higher with polymer precipitation [76]. Recently, Jia et al. reported that the PEG-based method isolated more EVs, proteins, and RNA than the UC method [77].

Ultrafiltration is a size method that is used to isolate EVs. It employs membrane filters with different pore sizes to allow smaller particles to penetrate and to pass through the membrane, while larger particles are excluded. Depending on the driving force, ultrafiltration can be classified as electric charge, centrifugation, and pressure. This method is efficient and simple, and allows for high-purity exosome isolation [78]. Ultrafiltration is a time- and cost-effective alternative to the gold-standard UC method [79]. Indeed, this EV isolation method is 50 times more efficient, and it removes smaller-sized proteins from EV suspension. However, one of the disadvantages is membrane pore blockage leading to low EV yield. Lamparski et al. demonstrated for the first time the possibility of isolating EVs for clinical application using the association of ultrafiltration and density gradient UC [80].

Size-exclusion chromatography (SEC) is an isolation process that is based on EV size. This column chromatographic approach offers a quicker method of vesicle enrichment and better standardization using commercially available columns, as recently highlighted by Böing et al. [81]. Guan Sheng et al. compared SEC and UC, and demonstrated that the recovery rate, structural integrity, and biological activity of EVs isolated using SEC were higher than those isolated via UC [82]. Moreover, the EV purity obtained is sufficient for proteomic and functional analyses [83]. However, the harvested EVs are severely diluted, and the elution processes are time-consuming. Nevertheless, no specific equipment is needed. UC and SEC methods could be used together for large-scale clinical applications such as drug delivery purposes.

Immunoselection is based on specific interactions between EV membrane proteins and the corresponding antibodies, allowing for EV separation from other molecules. Lipids, proteins, and polysaccharides are exposed on the EV surface, and they are thus potential ligands for selection. Antibodies to these surface proteins bind specific targeted EV populations via positive selection, and they remove irrelevant EVs, allowing for the isolation of a specific subclass of EVs [84]. The most commonly used targets are tetraspanins (CD9, CD63, and CD81), which allow for the isolation of total EVs [85]. Antibodies can be covalently attached to plates, beads, filters, or other matrices. This method requires a small number of samples, allows for the isolation of EVs with high purity, and induces no modifications in structure and morphology, but it is not adapted for clinical applications, and the cost of immunoselection is high.

These EV purification methods are not always mutually exclusive, and they can be combined to enhance the effectiveness of isolation and purification. Indeed, UC that is used to enrich EVs can be followed by SEC to remove proteins and contaminants [86].

**Table 1 pharmaceutics-15-00558-t001:** Principal EVs isolation methods, advantages, and disadvantages.

Methods	Mechanisms	Advantages	Disadvantages	References
Ultracentrifugation	Gold standard method based on sedimentation coefficient. Several centrifugation steps.	Large number of EVs with high purity. Simple and low cost. Isolation from large volumes.	Expensive. Specific infrastructure needed. Time-consuming. Low recovery rate. Potential EV damage.	[79,87]
Density gradients	Separation based on EV density and size.	Highly purified EVs. Preserved integrity.	Multistep procedure. Complex. Costly and time-consuming. Potential EV aggregation.	[72,88]
Polymer-based precipitation	Changes in EV solubility and aggregation using water-excluding polymers.	Fast and easy to use. Minimal cost. Suitable for large volume. No specialized equipment.	Low purity. Residual polymers and coprecipitation of contaminants.	[76,89]
Ultrafiltration	Size-based method. Membrane filters with specific size exclusion limits.	Simple and efficient. High purity and high productivity. No volume limitation. Can be associated with other methods.	EV deformation. Filter plugging. Loss of EVs via membrane attachment. Protein contamination.	[90,91]
Size-exclusion-based chromatography	Size-based separation. Columns filled with polymers with heterogeneous pores.	Simple and efficient. High purity and high quality. No EV damage. Separation of large and small molecules.	Long running time. Costly. Limitations on sample volume. Needs further enrichment.	[82,92]
Immunoaffinity	Based on specific interactions between immobilized antibodies and ligands on the EV surface.	High purity. High specificity. Isolation of EV subtypes. High recovery and good integrity. Can be combined with other methods.	High reagent costs. Not for large-scale purification.	[85,93]

## 6. Methods for Loading Drugs into EVs

A major challenge in applying EVs to DDS is to achieve an efficient loading of drugs/molecules into EVs [63]. Many different loading methods (Table 2), either endogenous or exogenous approaches, have been developed to promote the EVs drug delivery. The endogenous approach involves the modification of parental cells through transfection [9], lipofection [10], or coincubation with a drug, before the purification of these modified MSC-derived EVs [11]. Exogenous methods consist of loading drugs or molecules in EVs after their purification.

Several parameters may influence the incorporation of drugs into EVs: the structure of the EV, the drug properties, and the ratio of EV/drug. A wide range of drugs with different molecular weights can be loaded into EVs, but the choice of the loading method and the efficiency of encapsulation are very dependent on the properties of the drugs, in terms of their relative hydrophilicities/hydrophobicities [94,95]. Importantly, the analysis of physicochemical features, morphological appearance, and cellular uptake demonstrated that the EV integrity can be affected by the loading method [96]. The mean diameter of loaded EVs mostly increased, but the alteration was dependent on the drug properties, and the loading method confirming that the EV characterization before and after drug loading is essential [97].

Drugs can be coincubated with EVs, and they diffuse into the EVs along a concentration gradient. This passive method allows for the loading of hydrophobic drugs interacting with the lipid membrane of EVs. Through this method, small-molecule drugs such as curcumin [58], paclitaxel [98], and doxorubicin [99] have been effectively loaded in EVs. This method is simple, requires no additional stimulation, and preserves EV integrity, but the drug loading efficiency is low and is dependent on the hydrophobicity of molecules [100]. The efficiency can be increased by optimizing the incubation temperature, time, volume of buffer, and EV ratio.

EV donor cells can be treated with drugs (small molecules such as paclitaxel or doxorubicin and different types of RNA) to obtain drug-loaded EVs [101]. Two methods can be applied: transfection and coincubation. These allow for the loading of both hydrophilic and hydrophobic molecules. The efficiency of drug packaging into EVs depends on its concentration inside cells [102]. Interestingly, cells can be exposed to ultraviolet light and/or heat to stimulate the formation of drug-loaded EVs.

The exogenous method refers to the artificial incorporation of therapeutic molecules within EVs after their isolation and purification. This active loading requires EV permeabilization via different methods, such as electroporation, sonication, extrusion, freeze/thaw cycles, hypotonic dialysis, chemical methods, and incubation with membrane permeabilizers [103]. The molecules’ physiochemical properties determine which method best fits their encapsulation in EVs.

Electroporation is a simple and fast method. It consists of creating temporary small pores in the EV membrane under the action of an electric field, which increases membrane permeability. Molecules enter EVs through diffusion, and the membrane quickly recovers its integrity after drug loading. Nevertheless, electroporation requires specific equipment and the testing of optimum working conditions before the experiment. This approach has been used for loading EVs with curcumin or paclitaxel [58,104], and for encapsulating siRNAs or miRNAs. Some studies have shown that RNA and EVs can aggregate, resulting in a low loading capacity [105]. However, Bendix Johnsen et al. successfully optimized the use of a trehalose-containing buffer as a way of maintaining the structural integrity of EVs [106]. Recently, Liang et al. loaded a microRNA-21 inhibitor and chemotherapy drugs in EVs via electroporation [107].

The ultrasound method or sonication involves multiple ultrasonic treatments of a mix of EVs with the intended cargo. The mechanical shear force produced using the ultrasound probe compromises the integrity of the EV membrane and allows for drug encapsulation. This method is often used to load chemotherapeutic drugs in EVs [98], and provides superior drug loading compared with electroporation or coincubation. However, during sonication, a drug may adhere to the outer membrane layer, affecting its release. Some disadvantages of sonication include membrane integrity destruction and stability.

Mechanical extrusion can be used to encapsulate chemotherapeutic drugs. EVs are mixed with a drug, and the mixture is loaded into a syringe-based lipid extruder with 100–400 nm porous membranes under a controlled temperature. During extrusion, the EV membrane is disrupted and is vigorously mixed with the drug. This method allows for a high cargo loading efficiency, but this intensive extrusion process can change the EV membrane properties [15].

A freeze/thaw cycle strategy can be used to load drugs into EVs. It is a simple process [14]. Drugs are incubated with EVs at room temperature (RT) for a fixed amount of time, and then by performing at least three cycles of rapid freeze/thawing (−80 °C or in liquid nitrogen/RT), efficient EV drug loading is obtained. This method can induce EV aggregation, and the particle size increase. Moreover, the drug loading efficiency is lower than that obtained with sonication or extrusion [14]. Lee et al. used this method to prepare EVs containing miR-140 [108]. Recently, this strategy was used to create exosome-mimetic particles via membrane fusion between exosomes and liposomes [109].

Another approach for loading drugs into EVs is a hypotonic dialysis method based on the formation of drug transmembrane channels using osmotic pressure. This hypotonic environment allows for the penetration of small molecular substances via the opening of membrane pores. Fuhrmann et al. reported that porphyrin transfer in EVs can be drastically increased through hypotonic dialysis [15]. However, this method may induce EV size and charge changes.

Saponin has been described as an efficient permeabilization agent for cellular plasma membranes. Saponin can also increase the loading of different cargos in EVs. It creates pores in EV lipid bilayers through selective cholesterol removal. The EV loading of a small hydrophilic molecule, porphyrin, with saponin, allowed for an increased degree of loading (11-fold), in comparison with passive loading [15]. When compared with the simple incubation method, saponin was shown to enhance the loading of catalase into EVs and to preserve its activity [110].

Different methods result in varying loading efficiencies for the delivery of the same cargo. Chen et al. evaluated six commonly used drug-loading strategies (coincubation, electroporation, extrusion, freeze/thawing, sonication, and surfactant treatment) to incorporate Doxo into EVs at the single-particle level via nanoflow cytometry. The authors observed that the Doxo-loaded EVs prepared via coincubation and electroporation possessed a higher encapsulation ratio and a greater Doxo content than the EVs loaded with a single method. These Doxo-loaded EVs prepared via these two procedures resulted in a higher level of cellular uptake and induced more significant apoptosis for tumour cells, compared with EVs prepared with other drug-loading strategies [111].

**Table 2 pharmaceutics-15-00558-t002:** Principal methods for loading cargo in EVs after their isolation: advantages and disadvantages.

Methods	Mechanisms	Advantages	Disadvantages	References
Electroporation	Creation of pores under short and high voltage pulses	Wide applicability Simple and fast method RNAs and hydrophilic compounds	Aggregation Low loading capacity Morphological changes Special equipment	[58,104,105]
Sonication	Mechanical shear force produced using ultrasound probe compromises the integrity of the EV membrane, which permits drug encapsulation	High loading capacity Applicable for small RNAs	Destruction of membrane integrity Potential drug adhesion to the membrane affecting release	[98]
Extrusion	EVs are mixed with a drug and the mixture is loaded into a syringe-based lipid extruder with 100–400 nm porous membranes under a controlled temperature	High loading efficiency	Changes in EV membrane properties	[15]
Freeze/thaw cycles	Drug are incubated with EVs and at least 3 cycles of freeze/thawing (using −80 °C or liquid nitrogen)	Simple to perform Lower loading than sonication or extrusion Potential membrane fusion	EV aggregation Size increase	[14,109]
Saponin treatment	Pore formation in EV lipid bilayers via removal of cholesterol	High loading efficiency	Toxicity Loss of membrane integrity	[15,110]
Dialysis	Formation of drug transmembrane channels using osmotic pressure	Small molecular substances High drug-loading efficiency	EV size and charge changes	[15]

## 7. Loaded EVs for Therapy in Preclinical Studies

EVs have a great potential for drug delivery [5]. Different EV isolation protocols and loading methods have been tested and compared. In 2013, the first preclinical study used MSC-derived EVs transfected with miR-146 to treat glioma in an animal model [112]. The authors observed that MSC-derived EVs loaded with miR-146 elicited an antitumour effect in the rat brain. The authors demonstrated that the miRNA could be loaded into extracellular EVs, and that the plasmid-expressed miRNA was efficiently packaged into MSC-derived EVs via endogenous mechanisms. These findings suggest that miR-146b, delivered via MSC-derived EVs, is functionally active in acceptor tumour cells. Since then, numerous preclinical studies, both in vivo and in vitro, have indicated that MSC-derived EVs are safe and well tolerated. However, standardization is still lacking. A majority (61%) of the studies have used differential centrifugation and UC as the gold standard method, with or without a sucrose gradient, or under GMP production conditions. A commercial kit, with all types combined, was used in 39% of the studies (Figure 3). The electroporation loading method and plasmid transfection were the principal loading methods. The other loading methods were used more rarely (i.e., incubation or lentivirus transfection). There is great interest in MSC-derived EVs for oncological diseases and other conditions. MSC-derived EVs have been used in applications for glioma, breast cancer, melanoma, pancreatic cancer, hepatocellular carcinoma, colorectal cancer, acute myocardial infarction, myocardial ischemia–reperfusion injury, cerebral ischemia, and inflammatory diseases.

### 7.1. In Vitro Studies

In vitro, there have been many preclinical studies of EV-based therapy for pathological conditions, especially those in oncology. These studies aim to improve the treatment of these diseases using MSC-derived EVs loaded with proteins, miRNA, and drugs. The beneficial results have led to the development of most of the in vivo preclinical studies described in the next paragraph. Several studies have focused on fibrosis, ischemia, and inflammation reduction, but also on an increase in proliferation, migration, neurogenesis, and ageing prevention, through miRNA regulation mediated via EVs in cell lines [113,114]. Other studies have described that EVs have the ability to improve endothelial cell remodeling and angiogenesis, or decrease ischemia when cell lines are treated with EVs carrying specific proteins or drugs [115,116,117]. These studies support the emerging role of EVs as a drug delivery system (Table 3a).

Exosome-loaded miRNA (Exo-miRNA) or exosome-loaded drugs can act as a better delivery system, enhancing their effects on cancer cells. Li and colleagues reported on the synergistic effect of chemo-phototherapy by treating glioma cell lines with Exos CUR + ICG, combined with photothermal NIR radiation [113]. Katakowski et al. and Lang et al., respectively, observed the positive effects of Exo-miR-124 and Exo-miR-146 on glioma cells [112,118]. Melzer et al. reported that EVs loaded with paclitaxel (PTX) showed interesting outcomes in the treatment of breast cancer, with more efficient tumour-targeting properties [119], bonding receptor activity [120], and transcriptional regulation [121]. There are multiple benefits of EVs as nanocarriers in therapy, and Bagheri et al. reported on their potential to inhibit tumour growth in vitro [122]. Interestingly, Lou et al. described an increase in hepatocellular carcinoma (HCC) cell sensitivity to EVs containing doxorubicin, in comparison to free drugs [123]. Moreover, Yang et al. demonstrated interesting achievements in targeted therapy mediated by encapsulating doxorubicin in desialylated MSC-derived EVs on HCC cell lines [124]. Tumour growth inhibition induced by the apoptotic pathway can also be initiated by exosomes derived from TNFα plasmid-transfected MSCs [125]. The apoptotic pathway seems to play a key role when B16F0 melanoma cell lines are treated with Exo-TRAIL [126]. Cancer cell migration and proliferation are major points to be elucidated for a better understanding of oncology. Exosomes incubated with doxorubicin, recently described by Wei and colleagues, suppressed the migration and proliferation of osteosarcoma cells [117]. The last cancer in preclinical study is pancreatic cancer. This cancer is very resistant, and is considered as being almost “undruggable”, due to intense fibrosis and the immunosuppression of the TME [127]. Recent studies have shown promising results by targeting the specific mutation, KRAS^G12D^ that is involved in pancreatic ductal adenocarcinoma (PDAC). Both Mendt and collaborators, and Kamerkar and colleagues have shown that the use of engineered exosomes derived from MSCs in Panc-1 cells resulted in the upregulation of genes associated with lysosome, proteasome, or phagosome pathways, and in cell death [128,129]. Moreover, a better efficiency was observed, due to higher phosphorylated-ERK protein levels in Panc-1 cells treated with iExo sh/siRNA-KRAS^G12D^. In parallel, Ding et al. demonstrated E-cadherin and Bax upregulation, and Smad3, Bcl-2, and N-cadherin downregulation when pancreatic cells were treated with EV-miR-145 [130]. The following diagrams summarize the main effect in vitro (Figure 4).

**Table 3 pharmaceutics-15-00558-t003:** (**a**) Pre-clinical in vitro studies using EV as DDS in cancerous pathologies. (**b**) Pre-clinical in vitro studies using EVs as DDS in non-cancerous pathologies.

Diseases	Cell Lines	EV Sources	Active Pharmaceutical Ingredient (API)	API Loading Method (before/after EV Isolation)	Main Results	References
(**a**)						
Glioma	U87MG C6 HEB	BM-MSCs	Indocyanine green and curcumin	Electroporation (after isolation)	Exos-CUR + ICG caused cell inhibition by inducing apoptosis and cell arrest in G2/M phase, while a NIR-induced photothermal effect was synergistic with chemo-phototherapy, directly causing cell necrosis to achieve superior anticancer effects.	[113]
Glioma	GSC267 GSC20 GSC6-27 GSC8-11 GSC2-14	MSCs	miRNA-124a and PTEN-mRNA	Transfection (plasmid-based/before isolation)	Exo-miR124 reduced the viability and clonogenicity of GSCs compared with controls.	[118]
Glioblastoma multiforme	9L	MSCs	miRNA-146b	Transfection (plasmid-based/before isolation)	9L glioma cells treated by M146-exo showed a decrease in EGFR and NF-kB protein levels.	[112]
Breast cancer	TUBO 4T1	BM-MSCs	Doxorubicin	Electroporation (after isolation)	More efficient binding of LAMP2b-DARP in protein-exosomes to HER2-positive TUBO cells was observed, compared to HER2-negative 4T1 cells.	[120]
Breast cancer	A549 SK-OV3 MDA-hyb1	MSCs	Paclitaxel	Incubation (before isolation)	More efficient tumour-targeting properties were observed with drug-loaded Exos.	[119]
Breast cancer	TUBO 4T1	MSCs	miRNA-142-3p	Electroporation (after isolation)	anti-miR-142-3p-loaded Exos reduced the miR-142-3p and miR-150 levels, and increased the transcription of APC and P2X7R.	[121]
Colorectal cancer	MCF7 C26	MSCs	Doxorubicin	Electroporation (after isolation)	DOXO@exosomes-apt suppressed C26 and MCF7 cell growth.	[122]
Hepatocellular carcinoma	HCC Huh7 SMMC-7721 PLC/PRF HL-7702	MSCs	miRNA-199a	Transfection (lentivirus-based/before isolation)	Exo-199a delivery to HCC cells sensitized them to doxorubicin by targeting and inhibiting the mTOR pathway.	[123]
Hepatocellular carcinoma	HepG2	MSCs	Doxorubicin	Ultrasonication (after isolation)	Doxorubicin loaded in desialylated MSC-derived EVs as drug delivery system to target hepatoma cell lines.	[124]
Melanoma	MCF7 A549 Colo201 HCM HUVEC HKC L929	MSCs	TNF-α	Transfection (plasmid-based/before isolation)	CTNF-α-exosome-SPIONs enhanced tumour cell growth inhibition via the TNFR I-mediated apoptotic pathway.	[125]
Melanoma	B16F0	MSCs	TRAIL protein	Transfection (plasmid-based/before isolation)	Exo-TRAIL induced 2.5× more cell death (apoptosis level) compared to exosomes from non-treated B16F0 cells.	[126]
Osteosarcoma	MG63 HOS 143B H9C2	MSCs	Doxorubicin	Incubation (after isolation)	Osteosarcoma cell proliferation and migration were suppressed by Exo-Doxo.	[117]
Pancreatic cancer	PANC1 BxPC3	BM-MSCs	siKRAS^G12D^	Electroporation (after isolation)	siKrasG12D iExo upregulated genes associated with proteasome, lysosome, and phagosome pathways in Panc-1 cells.	[129]
Pancreatic cancer	PANC1 BxPC3 MIA-Capa21 Capan1	BM-MSCs	siKRASG12D and pLKO.1-shKRASG12D	Electroporation (after isolation)	KRASG12D mRNA and phosphorylated-ERK protein levels were reduced by iExosomes (with siRNA or shRNA targeting KRASG12D) in human Panc-1 cells.	[128]
Pancreatic cancer	HPDEC Capan1 CFPAC-1 BxPC3	hucMSCs	miRNA-145-5p	Transfection reagent (after isolation)	145-exo treatment resulted in the downregulation of Smad3, N-cadherin and Bcl-2 expression and upregulation of the E-cadherin and Bax genes in PDAC cells.	[130]
(**b**)						
Acute myocardial infarction	H9C2 EPCs	ADSC	miRNA-126	Transfection (miRNA-based/before isolation)	miR-126-exosomes prevented myocardial damage from inflammation, apoptosis, or fibrosis, and promoted angiogenesis.	[114]
Acute myocardial infarction	H9C2 EAhy926	MSCs	Akt	Transfection (adenovirus-based/before isolation)	Endothelial cell proliferation, migration, and tube-like structure formation were promoted by Akt-Exo.	[131]
Acute myocardial infarction	CFs H9C2 HUVEC	uc-MSCs	TIMP2 protein	Transfection (lentivirus-based/before isolation)	Exosomes derived from TIMP2-modified ucMSCs repaired the ischemia injuries by inhibiting apoptosis and promoting angiogenesis, and ECM remodeling in cardiomyocytes.	[116]
Acute myocardial infarction	Myocardial and endothelial cells (“homemade” isolation)	MSCs	Stromal-derived factor 1 (SDF1)	Transfection (plasmid-based/before isolation)	Autophagy and apoptosis were inhibited in myocardial cells via SDF1 overexpression mediated by EVs. Moreover, EVs promoted the microvascular regeneration of cardiac endothelial cells.	[115]
Myocardial ischemia reperfusion injury	Cardiomyocytes (“homemade” isolation)	BMSCs	miRNA-125b	Transfection (miRNA-based/before isolation)	I/R myocardium cells treated with BMSC-Exo-125b showed inhibition of apoptosis and inflammation, and an increase in cell viability.	[132]
Cerebral ischemia	BV-2	MSCs	miRNA-223-3p	Transfection (lentivirus-based/before isolation)	Exosomal miR-223-3p increased M2 microglia transformation into M1 microglia induced by NMLTC4 in a concentration-dependent manner, and decreased mRNA and protein expression of CysLT2R.	[133]
Cerebral ischemia	HeLa U87	BMSCs	Curcumin	Incubation (after isolation)	cRGD-Exo exhibited high affinity/specificity to cells expressing integrin avb3.	[134]
Ageing-induced vascular dysfunction	H9C2	ucMSCs	miRNA-675	Transfection (miRNA-based/before isolation)	miR-675 delivered by exosomes inhibited cell senescence. miR-675 mimic could inhibit ageing-related β-gal staining and promote cell proliferation in ageing cardiomyocytes.	[135]
Osteoarthritis	Chondrocytes (“homemade” isolated)	SMSCs	miRNA-140-5p	Transfection (lentivirus-based/before isolation)	The proliferation and migration of ACs were enhanced by SMSC-140-Exos without damaging ECM secretion.	[136]
Rheumatoid arthritis	HUVEC	MSCs	miRNA-150-5p	Transfection (plasmid-based/before isolation)	Exo-150 downregulated tube formation of HUVECs via MMP14 and VEGF pathways.	[137]
Intestinal fibrosis	IEC-6	BMSCs	miRNA-200b	Transfection (lentivirus-based/before isolation)	MiR-200b-MVs reversed the morphology in TGF-β1-treated IEC-6 cells.	[138]

MSCs: mesenchymal stromal cells; BMSC and BM-MSCs: bone marrow—mesenchymal stromal cells; SMSCs: synovial mesenchymal stromal cells; UC-MSCs/hucMSC/UMSCs: umbilical cord—mesenchymal stromal cells; GSC: glioblastoma stem cells; MDA-hyb1: breast cancer cells; AMSC: adipose mesenchymal stromal cells; AMI: acute myocardial infarction; ECM: extracellular matrix; NMLTC4: N-methyl LTC4 analogue; MI: myocardial infarction; SPION: superparamagnetic iron oxide nanoparticles; CTNF-α: cell-penetrating peptide coupled with TNF-α; cRGD: cyclo(Arg-Gly-Asp-D-Tyr-Lys) peptide; NDMAR1: N-methyl-D-aspartate receptor 1; OA: osteoarthritis; ACs: articular chondrocytes; CIA: collagen-induced arthritis; RVG: rabies virus glycoprotein; MVs: microvesicles; iExo: interfering exosomes.

Interesting reports have demonstrated the added value of EVs as nanocarriers in the therapy of non-oncological conditions (Table 3b). These studies have focused on acute myocardial infarction, myocardial ischemia–reperfusion injury, cerebral ischemia, or inflammatory diseases such as intestinal fibrosis, rheumatoid, and osteoarthritis, as well as vascular dysfunction that is ageing-associated [7]. The in vitro studies have shown encouraging results in all of these different conditions. Luo et al. described the preventive effect of Exo-miR-126 on myocardial damage [114]. Ma and colleagues reported that endothelial cell proliferation was promoted by Exo-Akt loading, which affected different signaling pathways [131]. Moreover, ischemia injury was repaired by the treatment of cardiomyocytes with EVs loaded with TIMP2 protein [116]. Indeed, microvascular regeneration was obtained via the overexpression of SDF-1 mediated by the EV preparation [115]. Similar results in cerebral ischemia and ageing-induced vascular dysfunction were reported by Zhao et al., Tian et al., and Tao et al. [133,134,136]. Encouraging studies on inflammatory diseases have shown improvements through the use of EV-based therapy. Groups studying osteoarthritis and rheumatoid arthritis have established the key role of miRNAs in vitro: miR-140-5p promoted the proliferation and migration of osteoarthritis cells treated with SMSC-140-Exos, and miR-150-5p encapsulated in exosomes led to a downregulation of tube formation by HUVECs via the VEGF and MMP4 signaling pathways [136,137,138].The following diagrams summarize (Figure 5) the main active pharmaceutical ingredients (API) in EVs derived from MSCs as drug delivery systems, and their effects in preclinical studies in the oncological and non-oncological fields.

### 7.2. In Vivo Animal Studies

Li et al. have shown in a mouse model that exos-based therapy can significantly abrogate glioma. Exos-based therapy consists of EVs loaded via electroporation with curcumin and indocyanine green [113]. Lang et al. showed the growth inhibition of glioblastoma stem cells (GSCs), also in an animal model, after treatment with EVs loaded with a supraphysiological level of miR-142a by plasmid transfection before EV isolation [118]. Interestingly, Katakowski and collaborators reported on a tumour growth decrease after rat treatment with EVs derived from MSCs transfected with a miR-146b plasmid [112]. In breast cancer, chemotherapeutic molecules such as paclitaxel or doxorubicin are loaded in EVs through incubation and electroporation, respectively. Gomari et al. observed a better inhibition of tumour growth with drug-loaded EVs than with free drugs [120]. Melzer et al. reported a 60% reduction in the subcutaneous primary tumour and distant organ metastases in mice with MDA-hy1 breast cancer after treatment with EVs incubated with paclitaxel [119]. In the same context, Naseri and colleagues reduced the expression levels of miR-142-3p and miR-150 associated with cancer stem cell tumourigenicity via the electroporation of LNA-antimiR-142-3p-loaded exosomes [121]. In the study of Bagheri et al., a single intravenous (IV) injection of EVs-DOXO significantly suppressed tumour growth in a mouse model of colorectal cancer [122]. In orthotopic mouse models of hepatocellular carcinoma (HCC), the IV injection of EVs-miR-199a loaded by the lentivirus transfection of adipose-tissue derived MSCs improved HCC chemosensitivity to doxorubicin via mTOR pathway targeting [123]. Yang et al. showed enhanced cytotoxicity efficiency using Doxo-loaded desialylated MSC-derived EVs for better targeting efficiency in Balb/c nude mice injected with HCC cells [124]. TNF-α and TRAIL protein can also be transfected using a plasmid in MSCs before EV isolation to obtain better drug delivery with increased antitumour activity, and less toxicity, in a melanoma mouse model [125]. Moreover, EXO-TRAIL reduced tumour progression by enhancing necrosis [126]. In pancreatic cancer, the deadliest disease, with a five-year overall survival rate of only 11%, Mendt and Kamerkar and their respective colleagues reported that EVs electroporated with siRNA-KRASG12D induced a robust antitumour effect in the PDAC model, and in multiple pancreatic cancer mouse models [128,129]. Kamerkar et al. reported on an increase in overall survival (OS) after the treatment of mice with EVs loaded with siRNA/pLKO.1-shRNA [128]. The following diagrams summarize the most relevant in vivo effects (Figure 6).

There has also been great interest in the therapeutic role of EVs in non-oncological conditions (Table 4b). In acute myocardial infarction (AMI), EVs loaded with miR-126, AKT, or TIMP2 protein via transfection allowed for recovery by decreasing the infarction area through a decrease in proinflammatory cytokines and a reduction in cardiac fibrosis [114,116,131]. Ma et al. treated AMI rats with AKT-loaded EVs, and observed an improvement in cardiac function through angiogenesis promotion [131]. Ni and colleagues demonstrated, in a rat model treated with exosomes derived from TIMP2-overexpressing MSCs, the apoptosis inhibition of cardiomyocytes and the promotion of angiogenesis and ECM remodeling in the context of AMI [116]. In a myocardial infarction (MI) mouse model, EVs collected from MSCs transfected with an SDF-1 plasmid inhibited cell autophagy and promoted microvascular endothelial cell production [115]. Many therapeutic effects of EVs as nanocarriers have been discovered in recent decades. Both in rat and mouse models, and in different pathological contexts, EVs display multiple therapeutic activities. In cerebral ischemia, miR-223-3p- or miR-124-loaded EVs have been used for ischemic cortex and hippocampus treatment in animal models, and they attenuate ischemia injury by stimulating neurogenesis [133,134,139]. Effects on senescence in ageing-induced vascular dysfunctions, chondrocyte proliferation in osteoarthritis, synovial inflammation, and joint damage in rheumatoid arthritis or oedema edema reduction in cerebral ischemia–reperfusion injury, attest to the great diversity of potential applications of EVs carrying miRNAs or molecules, bringing benefits for treatment and recovery from many diseases [135,137,140]. The following diagrams (Figure 7) summarize the main APIs in EVs derived from [135,137,140] MSCs as drug delivery systems, and their effects in preclinical studies in the fields of oncological and non-oncological diseases.

## 8. Loaded EVs for Therapy in Clinical Trials

More than 150 clinical studies involving EVs are in progress [141]. These studies aim to treat numerous conditions: bronchopulmonary dysplasia, dystrophic epidermolysis bullosa, acute ischemic stroke, chronic graft-versus-host diseases, macular holes, metastatic pancreatic cancer, and type I diabetes mellitus. However, only a few studies have been published. In 2014, the first patient suffering from steroid-refractory graft-versus-host disease (GvHD) was treated with hBMMSC-derived EVs [142]. The clinical GvHD symptoms improved briefly but significantly after the onset of MSC-derived EV therapy. The patient was stable for a few months. The obtained results suggested that MSC-derived EVs may provide a potentially new and safe tool to treat refractory GvHD and other inflammation-associated diseases. Nassar and colleagues also observed that EV therapy could ameliorate inflammatory immune reactions [143]. Interestingly, the authors demonstrated that the administration of cell-free cord-blood MSC-derived EVs was safe and could improve the inflammatory immune reaction and improve the overall kidney function in grade III-IV CKD patients.

Currently, no studies using MSC-derived EVs as nanocarriers have been published. Several preclinical studies have been published, which have shown optimistic results [144]. Munoz et al. demonstrated that the delivery of functional anti-miR-9 by MSC–derived EVs to glioblastoma multiforme cells conferred chemosensitivity [145]. Melzer et al. observed that the systemic intravenous application of Taxol-loaded MSC-derived EVs induced a greater than 60% reduction in subcutaneous breast tumours [119]. Moreover, the number of distant organ metastases observed in the lung, liver, spleen, and kidney was reduced by 50% with Taxol-loaded MSC-derived EVs, similar to the effects observed with Taxol. However, the Taxol concentration in EVs was reduced by approximately 1000-fold. Few clinical studies are in progress (Home-ClinicalTrials.gov). Kamerkar et al. evaluated the best dose and side effects of MSC-derived EVs with KrasG12D siRNA (iEVs) to treat metastatic pancreatic cancer patients harboring the KrasG12D mutation, in a phase I trial (NCT03608631) [146]. This clinical trial is in the recruitment phase, and 28 participants will be included. These patients will receive EVs IV over 15–20 min on Days 1, 4, and 10. The treatment will be repeated every 14 days for up to 3 courses. The study should be completed in March 2023.

MSC-derived EVs are also currently studied in non-oncological conditions such as homozygous familial hypercholesterolemia or acute ischemic stroke. The first trial (NCT05043181) aims to create an LDL receptor-expressing virus vector to generate LDLR mRNA-enriched EVs derived from BM-MSCs and purified via filtration and ultracentrifugation. EVs will be injected through abdominal puncture to evaluate their safety and efficacy for the treatment of homozygous familial hypercholesterolemia patients presenting a functional loss due to the mutation of the LDLR. Thirty patients will be included, and the study will be finalized in December 2026 [147]. The second study (NCT03384433) aims to assay the administration of MSC-derived EVs on the improvement of disability in patients with acute ischemic stroke. Five patients will receive allogenic MSC-derived exosomes transfected by miR-124, 1 month after the attack via stereotaxis/intraparenchymal injection. This randomized, single-blind, placebo-controlled phase I/II trial was supposed to be completed in December 2021, but no data have been published [148].

## 9. Challenges

Even though the results of preclinical studies have been positive, different steps are needed to overcome quality control and procedure standardization. Indeed, different protocols of EV purification, quantification, and characterization coexist [144]. The lack of standardized isolation and purification methods for EVs, the limited drug delivery efficiencies of EVs, the isolation of EVs contaminated with impurities (cell debris, non-exosomal vesicles, and proteins) and insufficient production are still major challenges. For drug delivery, the evaluation of storage conditions, pharmacokinetics, and the biodistribution of loaded EVs is needed. In addition, the culture of MSCs that produce EVs must also be considered. The bioreactors for cell expansion should provide sufficient EV quantities for clinical-grade production.

## Figures and Tables

**Figure 1 pharmaceutics-15-00558-f001:**
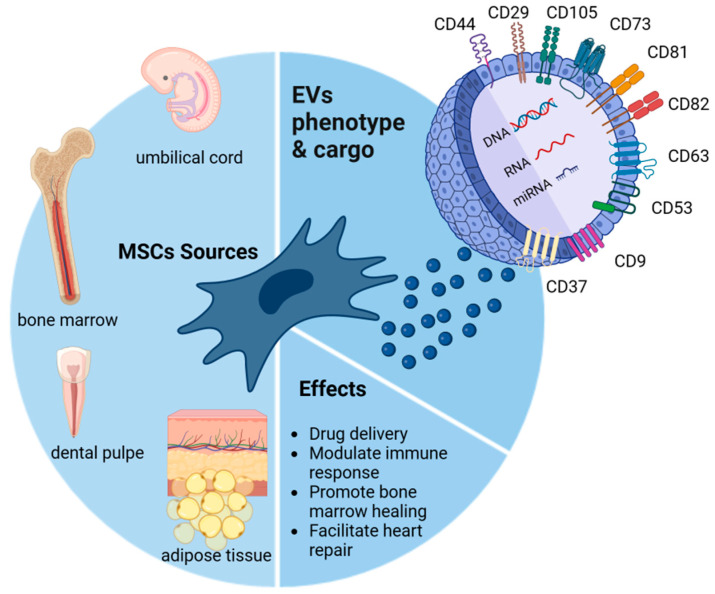
Essential mesenchymal stromal cell (MSC) sources and their extracellular vesicle (EV) phenotypes, cargos (CD29, CD44, CD73, CD105, CD81, CD82, CD63, CD53, CD9, CD37, DNA, RNA, and miRNA), and summarized effects.

**Figure 2 pharmaceutics-15-00558-f002:**
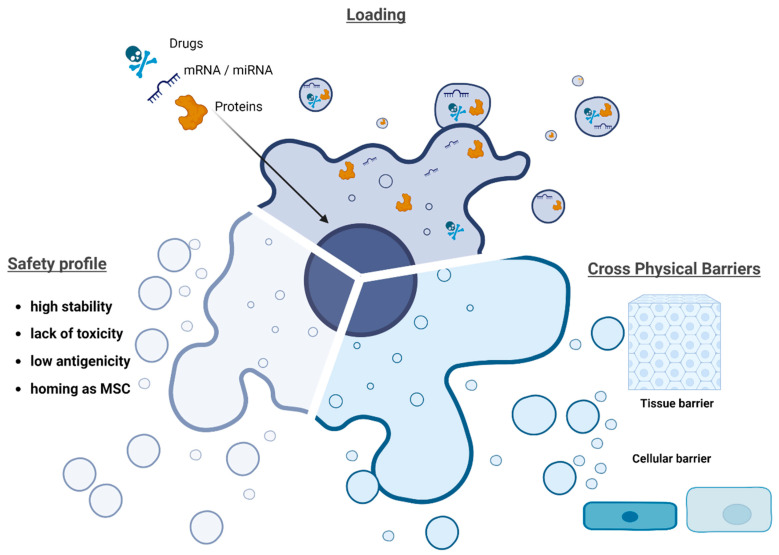
Advantages of extracellular vesicles (EVs) vs. liposomes. Loading cargos, safety profile, and their potential to cross barriers.

**Figure 3 pharmaceutics-15-00558-f003:**
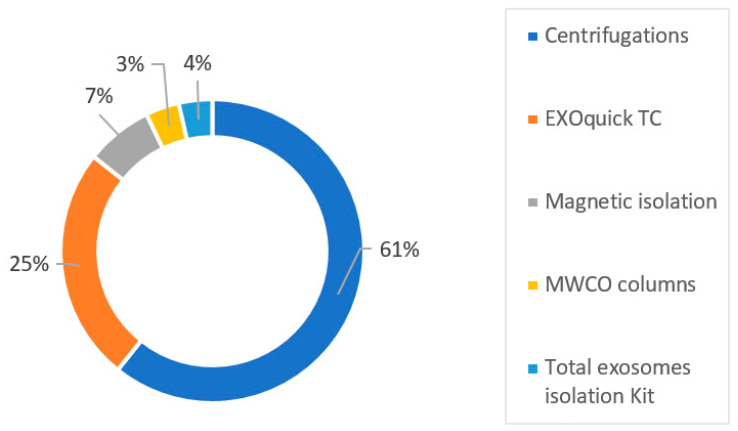
EV isolation methods in preclinical studies.

**Figure 4 pharmaceutics-15-00558-f004:**
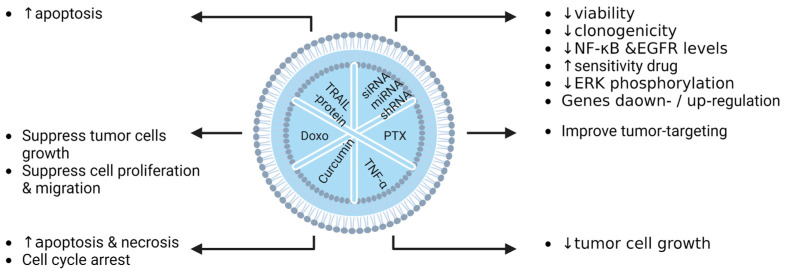
Summary of EV cargos, and their relevant effects on cancerous pathologies in vitro (Doxo: doxorubicin, PTX: paclitaxel, ↑: increase, ↓: decrease).

**Figure 5 pharmaceutics-15-00558-f005:**
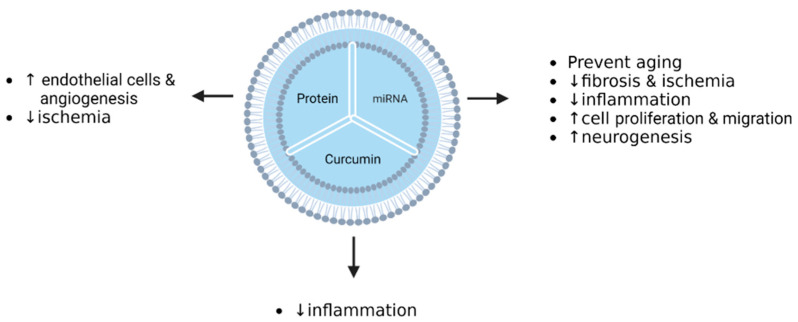
Summary of EV cargos and their effects on non-cancerous pathologies in vitro (↑: increase, ↓: decrease).

**Figure 6 pharmaceutics-15-00558-f006:**
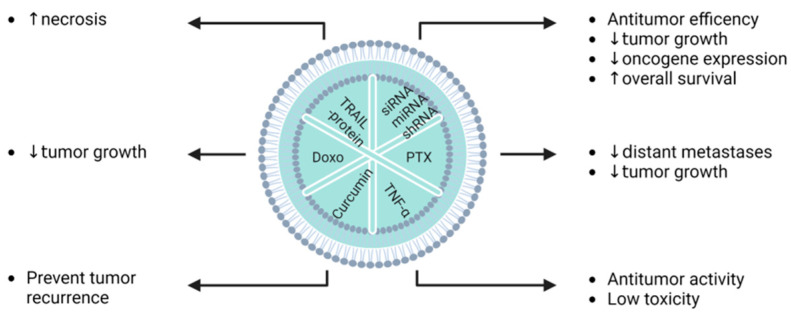
EV cargos and their in vivo effects on cancerous pathologies (Doxo: doxorubicin, PTX: paclitaxel, ↑: increase, ↓: decrease).

**Figure 7 pharmaceutics-15-00558-f007:**
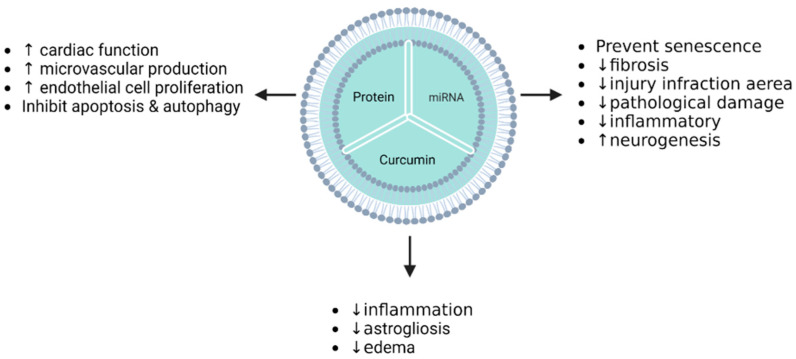
EV cargos and their in vivo effects on non-cancerous cells (↑: increase, ↓: decrease).

**Table 4 pharmaceutics-15-00558-t004:** (**a**) Pre-clinical in vivo studies using EV as DDS in cancerous pathologies. (**b**) Pre-clinical in vivo studies using EV as DDS in non-cancerous pathologies.

Disease	Model	EV Sources	Active Pharmaceutical Ingredient (API)	API Loading Method (before/after EVs Isolation)	Main Results	References
(**a**)						
Glioma	Mouse	BM-MSCs	Indocyanine green and curcumin	Electroporation (after isolation)	Exos-based combined therapy drastically abrogated glioma and increased the prevention of rapid tumour recurrence following transient phototherapy and total tumour remission in a mouse model.	[113]
Glioma	Mouse	MSCs	miRNA-124a and PTEN-mRNA	Transfection (plasmid-based/before isolation)	Exosomes loaded with a supraphysiological level of miR124a inhibited the growth of GSCs in mice with an intracranial GSC xenograft.	[118]
Glioblastoma multiforme	Rat	MSCs	miRNA-146b	Transfection (plasmid-based/before isolation)	The injection of miR146 expressing MSC-derived exosomes inside the tumour decreased the growth of tumours in rats with a glioma xenograft.	[112]
Breast cancer	Mouse	BM-MSCs	Doxorubicin	Electroporation (after isolation)	Targeted doxorubicin-loaded exosomes showed specific delivery to the target tissues in a murine breast cancer model, and reduced the tumour growth rate, compared to free drug or untargeted exosomes.	[120]
Breast cancer	Mouse	MSCs	Paclitaxel	Incubation (before isolation)	Systemic IV injection of MSC-derived Taxol exosomes reduced by 60% the subcutaneous primary tumour, and distant organ metastases in NODscid mice with metastatic MDA-hyb1 breast cancer.	[119]
Breast cancer	Mouse	MSCs	miRNA-142-3p	Electroporation (after isolation)	LNA-anti-miR-142-3p MSCs-derived exosomes reduced the expression level of miR-1423p and miR150 in tumour-bearing mice.	[121]
Colorectal cancer	Mouse	MSCs	Doxorubicin	Electroporation (after isolation)	Ectopic model of C26 in BALB mice showed that a single IV injection of targeted DOXO@exosomes-apt significantly suppressed the tumour growth compared to free DOXO.	[122]
Hepatocellular carcinoma	Mouse	MSCs	miRNA-199a	Transfection (lentivirus-based/before isolation)	AMSCs-Exo-199a could be used to distribute miR199a to tumour tissue. Moreover, they increased the chemotherapeutic effects of doxorubicin by targeting and inhibiting the mTOR pathway.	[123]
Hepatocellular carcinoma	Mouse	MSCs	Doxorubicin	Ultrasonication (after isolation)	Doxorubicin loaded in desialylated MSC-derived EVs as a drug delivery system targeted hepatoma cells in mouse model.	[124]
Melanoma	Mouse	MSCs	TNF-α	Transfection (plasmid-based/before isolation)	Coupled SPIONs and CTNF-α anchored exosomes delivered peptide drugs to the cytomembrane better than to the cytoplasm, and resulted in an increase in antitumour activity and lower toxicity.	[125]
Melanoma	Mouse	MSCs	TRAIL protein	Transfection (plasmid-based/before isolation)	Homing ability to Exo-TRAIL reduced tumour progression by enhancing necrosis in cancer cells following multidose administration in both in vivo and in vitro models.	[126]
Osteosarcoma	Mouse	MSCs	Doxorubicin	Incubation (after isolation)	Exo-DOXO displayed higher cytotoxicity than free drug, and was efficient as a drug delivery system.	[117]
Pancreatic cancer	Mouse	MSCs	siKRAS^G12D^	Electroporation (after isolation)	Both BM-MSCs- and BJ-MSCs-derived exosomes loaded with siKRAS^G12D^ showed a robust antitumour efficiency in PDAC models.	[129]
Pancreatic cancer	Mouse	MSCs	siKRASG12D and pLKO.1-shKRASG12D	Electroporation (after isolation)	Exosomes derived from mouse skin fibroblast were used as a nanocarrier to specifically target pancreatic cancer cells in multiple mouse models of pancreatic cancer. EV injection drastically increased OS.	[128]
Pancreatic cancer	Mouse	hucMSCs	miRNA-145-5p	Transfection reagent (after isolation)	Intratumour injection of miR145-5p UC-MSCs-derived exosomes reduced xenograft tumour growth in a BALB/c mouse model of Panc-1 cells.	[130]
(**b**)						
Acute myocardial infarction	Rat	Adipose stem cells	miRNA-126	Transfection (miRNA-based/before isolation)	The treatment of AMI rats with miR-126-enriched exosomes decreased the infarction area in myocardial injury, inflammatory cytokine expression, and cardiac fibrosis.	[114]
Acute myocardial infarction	Rat	MSCs	Akt	Transfection (adenovirus-based/before isolation)	Exosomes derived from Akt-modified hucMSCs promoted angiogenesis, in which PDGF-D was involved in Akt-Exo-mediated angiogenesis. Additionally, they improved cardiac function in rats with AMI induced by LAD ligation.	[131]
Acute myocardial infarction	Rat	MSCs	TIMP2 protein	Transfection (lentivirus-based/before isolation)	Exosomes derived from huc-MSCs via the Akt/Sfrp2 pathway inhibited apoptosis in cardiomyocytes and promoted angiogenesis and ECM remodeling in ischemic myocardium.	[116]
Acute myocardial infarction	Mouse	MSCs	Stromal-derived factor 1	Transfection (plasmid-based/before isolation)	Inhibition of ischemic myocardial cell autophagy and microvascular production of endothelial cells were promoted in MI mice treated with Exo-SDF10.	[115]
Myocardial ischemia–reperfusion injury	Rat	BMSCs	miRNA-125b	Transfection (miRNA-based/before isolation)	Injection of BM-MSCs-Exo-125b reduced pathological damage and decreased SIRT7 level expression in I/R rats model tissues.	[132]
Cerebral ischemia	Rat	MSCs	miRNA-223-3p	Transfection (lentivirus-based/before isolation)	Ischemic cortex and hippocampus MCAO/R surgery-mediated injury were treated by miR-223-3p-MSC-derived exosomes.	[133]
Cerebral ischemia	Mouse	BMSCs	Curcumin	Incubation (after isolation)	cRGD-Exo-cur suppressed inflammation by targeting NF-*κ*B.	[134]
Cerebral ischemia	Mouse	BM-MSCs	miRNA-124	Electroporation (after isolation)	Cortical neural progenitors were promoted by systemic administration of RVG-exosomes miR-124. Ischemia injury was attenuated by stimulating neurogenesis.	[139]
Cerebral ischemia–reperfusion injury	Mouse	MSCs	Curcumin	Incubation and freeze/thaw cycle (after isolation)	IR-injury mice treated by MESC-exo^cur^ showed a reduction in neurological score, oedema, astrogliosis, NDMAR1 expression, and inflammation.	[140]
Ageing-induced vascular dysfunction	Mouse	UMSCs	miRNA-675	Transfection (miRNA-based/before isolation)	Targeting the TGF-β1/p21 pathway by miR-675 UC-MSCs exosomes prevented senescence, ischemic legs, and muscle ageing.	[135]
Osteoarthritis	Rat	SMSCs	miRNA-140-5p	Transfection (lentivirus-based/before isolation)	OA rat model treated with sMSC-140-Exos showed delayed early-stage OA progression by promoting chondrocyte proliferation and migration via the inhibition of SOX9 and ECM.	[136]
Rheumatoid arthritis	Mouse	MSCs	miRNA-150-5p	Transfection (plasmid-based/before isolation)	Inhibition of MMP-14 and TNF, driven by Exo-150-5p, decreased synovial inflammatory and joint damage in a CIA mouse model.	[137]
Intestinal fibrosis	Rat	BMSCs	miRNA-200b	Transfection (lentivirus-based/before isolation)	EMT remodeling and the target protein ZEB1/2 alleviated colon fibrosis via treatment of a rat model with miR-200-MVs.	[138]

MSCs: mesenchymal stromal cells; BMSC and BM-MSCs: bone marrow and mesenchymal stromal cells; BJ-MSCs: foreskin fibroblasts-mesenchymal stromal cells; SMSCs: synovial mesenchymal stromal cells; UC-MSCs/hucMSC/UMSCs: umbilical cord/mesenchymal stromal cells; GSC: glioblastoma stem cells; IV: intravenous; NOD/SCID: nonobese diabetic severe combined immunodeficient mice; MDA-hyb1: breast cancer cells; AMSC: adipose mesenchymal stromal cells; AMI: acute myocardial infarction; ECM: extracellular matrix; LAD: left anterior descending coronary artery; OS: overall survival; MI: myocardial infarction; MCAO/R: middle cerebral artery occlusion/reperfusion; SPION: superparamagnetic iron oxide nanoparticles; CTNF-α: cell-penetrating peptide coupled with TNF-α; cRGD: cyclo(Arg-Gly-Asp-D-Tyr-Lys) peptide; NDMAR1: N-methyl-D-aspartate receptor 1; OA: osteoarthritis; CIA: collagen-induced arthritis; RVG: rabies virus glycoprotein; MVs: microvesicles; PDAC: pancreatic ductal adenocarcinoma cancer.

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
