# Peer review of "Potential of Mesenchymal Stromal Cell-Derived Extracellular Vesicles as Natural Nanocarriers: Concise Review"

_pharmaceutics, 2023, doi:10.3390/pharmaceutics15020558_

Round 1

Reviewer 1 Report

Therapeutic potential of mesenchymal stromal cellderived extracellular vesicles as natural nanocarriers: concise review.

The authors present a well written, scientifically sound review article that details the use of EVs as the next-generation nanocarries, biomarkers and therapeutic agents. 

Minor Revision

There are literature reviews published on this topic which showcase the use of EVs for delivery of drugs and therapeutics, the authors have not cited the recently published work in their article. They need to include and acknowledge the work that is already done and differentiate how their work is novel.

e.g. Extracellular vesicles as a next-generation drug delivery platform this review article from 2021. 

Reviewer 2 Report

In the manuscript by Draguet et al, “Therapeutic potential of mesenchymal stromal cellderived extracellular vesicles as natural nanocarriers: concise review.”. In this manuscript, authors tried to review the therapeutic potential of mesenchymal stromal cell-derived extracellular vesicles, however, the following comments/suggestions must be addressed before publication.

-There are a lot of typos and formatting error in the manuscript which should be addressed.

-Authors have did not clearly focus on one application of EVs (therapeutic agents or drug delivery system). This should be very clear. Without this, this is incomplete.

-Authors have presented the tables (isolation methods for EVs) which are already in the literature since several years. Same as drug loading.

-As title suggests as natural nanocarriers, did authors tried to check those surface cues which present on the EV surface and list them as natural targeting agents. Please collect information from literature and list them (proteins or nucleic acids).

-If authors are trying to explore the therapeutic potential of EVs, then need to explain the role of therapeutic cargos responsible for this.

Reviewer 3 Report

Dear Authors,

There are some minor comments and corrections 

Figure 1: Tetraspanin markers are not clearly visible and it will be helpful for the readers if tetraspanin markers, DNA, RNA are labelled in the figure.

Figure 3: Please mention the abbreviated techniques in the pie chart

Table 2, 3a, b and 4a, b - title is missing 

Please be consistent with DSS abbreviation. Both DSSs and DSS are mentioned in the text.

line 93: Placentae 

line 215: polyethylene - glycol is missing 

Reviewer 4 Report

In this manuscript the authors made a comprehensive review on mesenchymal stromal cell‐derived extracellular vesicles (EVs) as drug carriers. Different isolation methods and different drug loading techniques on different types of drugs were listed and discussed. Examples were made for such carriers in the applications of in vitro, in vivo and clinical studies. The manuscript may be of interesting for publication after the revisions by taking some arguments and suggestions into consideration.

1) Should the physical and chemical properties of EVs be included for introduction besides their applications? What are the structural advantages of EVs over other carriers such as phospholipids, nanoparticles and biopolymers?

2) Should the reason of choosing mesenchymal stromal cell‐derived EVs be discussed? Why mesenchymal stromal cells are of interest particularly? What are the advantages (uniqueness) of mammalian mesenchymal stromal cell‐derived EVs over other types of cell‐derived EVs such as bacteria, fungi and plant cells?

3) Should the importance of size, morphology and drug compatibility of the EVs be discussed when discussing drug encapsulation by EVs? These are important factors to be considered in the drug delivery system.

4) Figure 3 cannot be correlated well with the explanation in line 371 to 393. The meaning of percentage numbers in the figure are confusing. More figure labeling and clearer explanations are needed for better understanding.

Round 2

Reviewer 2 Report

Authors have addressed the comments and suggestions, it is recommended to accept.

Thanks!

Reviewer 4 Report

The authors' response has resolved the issues and questions in the original manuscript. please see attachment for details
